# Seasonal Response of Major Phytoplankton Groups to Environmental Variables along the Campeche Coast, Southern Gulf of Mexico

Juan Alfredo Gómez-Figueroa [1], Jaime Rendón-von Osten [1], Carlos Antonio Poot-Delgado [1,*], Ricardo Dzul-Caamal [1] and Yuri B. Okolodkov [2]

1   Instituto de Ecología, Pesquerías y Oceanografía del Golfo de México, Universidad Autónoma de Campeche (EPOMEX-UAC), Apdo. Postal 520, Campeche 24030, Mexico

2   Instituto de Ciencias Marinas y Pesquerías (ICIMAP-UV), Universidad Veracruzana, Calle Mar Mediterráneo Núm. 314, Fracc. Costa Verde, Boca del Río 9429, Mexico; yuriokolodkov@yahoo.com

*   Correspondence: cpoot35@gmail.com

**Abstract:** To describe the seasonal response of the major phytoplankton groups to environmental variables along the Campeche coast, southeastern Gulf of Mexico, seven shallow-water (ca. 1 m) stations were monitored from January 2019 to January 2020. Orthophosphate, ammonium, nitrite, nitrate and silicate were measured. Several tests, including ANOVA, the Kolmogorov–Smirnov test, Tukey TSD, Bartlett's test and canonical correspondence analysis (CCA), were applied. The physicochemical variables (temperature, salinity and pH) recorded are typical for the central coast of Campeche. Seasonal characteristics are affected by the shallowness of the study area. The variation of inorganic nutrient concentrations is likely to be related to specific polluting activities. While the abundance of phytoplankton presented a minimum value of $4.1 \times 10^4$ cells $L^{-1}$ in March, the maximum value of $8.8 \times 10^6$ cells $L^{-1}$ occurred in May; the general average was $5.3 \times 10^5$ cells $L^{-1}$. Based on CCA, the correlation between major phytoplankton groups and physical–chemical variables was high ($r \approx 0.8$), indicating a significant relationship. The CCA graphs separated the samples of diatoms by higher values of pH and silicate and separated the samples of cyanobacteria with high values of temperature (>30 °C) from the samples with dinoflagellates and nanoflagellates. Nanoflagellates were abundant in the samples with high values of ammonium and phosphate.

**Keywords:** annual cycle; cyanobacteria; diatoms; dinoflagellates; Gulf of Mexico; microalgae; nanoflagellates; nutrients; phytoplankton; seasonal changes

## 1. Introduction

Marine ecosystems are subject to anthropogenic disturbances such as climate change and eutrophication along coastal zones [1]. This causes a complex interaction between physical and chemical cycles, which harms many of the species of aquatic fauna that depend on abiotic and biotic conditions [2], and these disturbances can interfere with the structure and dynamics of phytoplankton communities [3] because they are closely related to the interaction of the aforementioned factors [4]. Phytoplankton communities have been extensively studied as bioindicators of water quality in different aquatic environments, relating physicochemical parameters of water quality, level of eutrophication and/or saprobity with the presence/absence, richness and abundance of major microalgal groups present in monitored sites, and are strongly linked to the nutrient stoichiometric balance [5]. According to Jørgensen et al. [6], the initial energy intake, growth and development of ecosystems are possibly due to an increase in biomass, energy flows and then in biodiversity. When diversity increases, feedback controls increase, effective specific respiration decreases and there is a tendency to substitute r-strategist species for k-strategists, meaning less energy is wasted on reproduction. Consequently, statistical analysis using species and

environmental data-based aggregation and classification could identify key factors for environmental variables in the marine environment [7].

The State of Campeche is on the southeastern Gulf of Mexico, and it is part of the five maritime states of the Mexican Gulf of Mexico. Its coastline is 425 km in length, which represents 3.8% of the total Mexican coastline. It ranks seventh in production and eighth in the value of fisheries and aquaculture production at the national level [8]. Furthermore, it is a place where basic information about human health is scarce.

For all the above, the objectives were to determine the responses of major phytoplankton groups by using statistical methods that allowed the identification of the relationships between water quality and phytoplankton groups, as well as the identification of possible effects of environmental variables.

## 2. Materials and Methods

The study area is located in the central part of the coast of the State of Campeche, with a depth of 4 m to less than 1 m [9], characterized by calcareous sediments that originated from the carbonate platform in Yucatán [10]. A tidal cycle is 28 days, with a tidal range of 0.1 to 0.9 m and mean values of 0.38–0.41 m [11].

Surface seawater samples were collected from January 2019 to January 2020, at seven stations with an approximate depth of 1 m. The sampling sites correspond to rainwater discharge points and are close to areas of anthropogenic activity located in the coastal waters of the State of Campeche (Figure 1). The sampling dates for the analyses were grouped according to the meteorological conditions previously reported for the area [12]: the dry season (February to May), the rainy season (June to September) and the windy season (October to January).

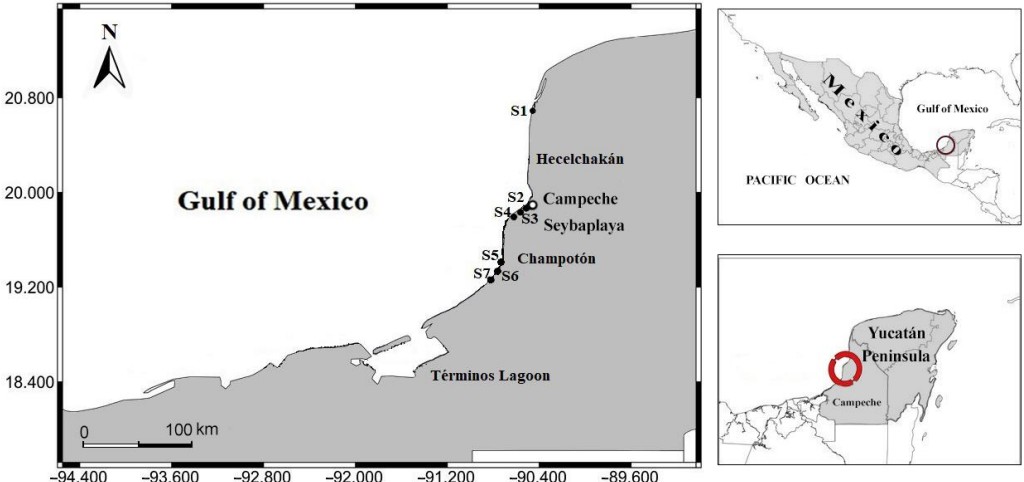

**Figure 1.** Sampling sites (S1–S7; marked by black circles) in the coastal waters of the State of Campeche. On the maps to the right, red circles indicate the study area.

Phytoplankton samples were taken using a plastic bottle (capacity 1 L), taking a 100 mL aliquot, to analyze abundance and phytoplankton composition. The samples were fixed in the field with a 1% neutral iodine solution and subsequently preserved by adding a 4% neutralized formalin solution [13]. At each station, using the multiparameter HANNA model HI9828 (Woonsocket, RI, USA), the following variables were measured at the site: water temperature (°C), salinity, pH, dissolved oxygen concentration (DO) and DO saturation (% DO).

The analyses of inorganic nutrients (nitrite, nitrate, ammonium, orthophosphate and silicate) were performed in the laboratory following standard chemical methods for marine environmental monitoring [14]. In this study, the physical–chemical variables measured in situ were conventionally separated from inorganic nutrients.

The quantification of the phytoplankton cells was carried out according to the Utermöhl technique [15], taking 10 cm$^3$ of the sample, which was deposited in a sedimentation chamber with a glass lid for 24 h, so that all the materials making up the sample settled. Quantification was performed using an IROSCOPE SI-PH (series 450, Mexico City, Mexico) phase-contrast inverted microscope with the 10×/0.25 Ph1 ADL and LD 25×/0.30 Ph1 objectives. It should be noted that within nanoflagellates (<20 μm), due to their size, phototrophic and heterotrophic species were not differentiated. The abundance values are expressed in cells per liter (cells L$^{-1}$). To identify phytoplankton taxa, specialized literature was consulted [16–21].

To determine whether the data on physical–chemical variables, inorganic nutrients and phytoplankton abundances presented a normal distribution, the Kolmogorov–Smirnov test and homoscedasticity with Bartlett's test were applied [22]. Differences between the sampling months and between the seasons were analyzed with ANOVA and Tukey TSD (Truly Significant Difference), with a significance level of 0.05 [23]. The calculation routine was performed with the Statgraphics Centurion XV program, version 18.2.06. SigmaPlot (version 10.0) was used for graphics.

A canonical correspondence analysis (CCA) was carried out to identify the possible effects of the set of physical–chemical variables and inorganic nutrients on the major groups of phytoplankton. Prior to the analysis, the data were transformed to Log 10 (data + 1) to reduce the difference in magnitude between the variables. Additionally, through the Monte Carlo analysis with 499 permutations, the importance of the axes of the CCA was tested [24]. The calculation routine was carried out using the CANOCO program, version 4.5.

## 3. Results

### 3.1. Physical–Chemical Variables

The study area was characterized by a temperature range of ±2 °C, with the average minimum values during the windy season from 23.70 to 26.10 °C. The average maximum values, ranging from 31.10 to 36.50 °C (Table 1), were recorded during the rainy season. The salinity mean values during the rainy season ranged from 28.60 to 36.00, and the minimum average values during the windy season ranged from 15.89 to 32.97 (Table 1). The maximum average pH levels were recorded during the rainy season with a range of 8.10 to 8.50, while the minimum values were registered during the dry season (7.88) and the windy season (7.99) (Table 1). The pH values were within the upper limits established for marine coastal waters. Maximum average DO values (6.59–15.08 mg L$^{-1}$) were observed in the windy season (Table 1), while minimum average values were recorded in the dry season (4.70–7.99 mg L$^{-1}$). Minimum values of DO were recorded during the rainy and dry seasons below those established for the limits for marine coastal waters. While minimum average oxygen saturation values were recorded in the dry season (64–111.10%), maximum average values (78.90–182.50%) were observed in the windy season (Table 1). All the above-mentioned variables showed significant differences among seasons ($p < 0.05$).

### 3.2. Inorganic Nutrients

The concentrations of inorganic nutrients varied significantly (Table 1). The maximum concentrations of nitrite (0.36 to 3.24 mg L$^{-1}$), nitrate (9.76 and 51 mg L$^{-1}$), ammonium (5.06–29.58 mg L$^{-1}$) and phosphate (14.86–57.44 mg L$^{-1}$) occurred during the rainy season. The concentrations of inorganic nutrients showed significant differences between seasons ($p < 0.05$). The variations in the concentrations of inorganic nutrients were within the upper limits established for marine coastal waters. Silicate levels showed maximum mean values (1.60–9.70 mg L$^{-1}$) in the rainy season, while minimum mean values (0.80–8.50 mg L$^{-1}$) were measured in the dry season. No significant differences were observed between seasons (F = 0.09; $p > 0.05$). It should be noted that silicates are not included in the current Mexican regulations.

**Table 1.** Statistics for meteorological seasons for physical–chemical variables and inorganic nutrients. Physical–chemical variables and inorganic nutrients obtained from the period 2019–2020 are presented as range and mean ± standard deviation.

| Season | Physical–Chemical Variables | | | | |
| --- | --- | --- | --- | --- | --- |
| | T $^\circ$C | Salinity | pH | DO (mg L$^{-1}$) | DO% |
| Rainy | 31.10–36.50 31.10–36.50 32.77 ± 1.85 | 28.60–36.00 31.98 ± 2.97 | 8.10–8.50 8.36 ± 0.16 | 4.86–11.99 7.73 ± 2.28 | 66.70–175.50 108.46 ± 35.17 |
| Dry | 30.80–31.90 31.54 ± 0.44 | 29.01–35.39 32.79 ± 2.59 | 7.50–8.27 7.88 ± 0.28 | 4.70–7.99 6.42 ± 1.32 | 64.00–111.10 89.66 ± 19.05 |
| Windy | 23.70–26.10 24.94 ± 0.84 | 15.89–32.97 22.56 ± 5.26 | 7.52–8.30 7.99 ± 0.24 | 6.59–15.08 11.63 ± 3.51 | 78.90–182.50 140.50 ± 43.23 |
| Differences between seasons | F = 88.41 $p < 0.005$ | F = 15.70 $p < 0.005$ | F = 8.08 $p < 0.005$ | F = 8.00 $p < 0.005$ | F = 4.00 $p < 0.005$ |
| * Upper limits established for marine coastal waters | ±1.5 of natural conditions reported previously | | >5 to >10 | >5 mg L$^{-1}$ | |

| Season | Inorganic Nutrients (mg L$^{-1}$) | | | | |
| --- | --- | --- | --- | --- | --- |
| | Nitrite | Nitrate | Ammonium | Phosphate | Silicate |
| Rainy | 0.38–3.24 0.99 ± 1.01 | 9.76–51.00 35.70 ± 17.56 | 5.06–29.58 10.54 ± 8.57 | 14.86–57.44 36.10 ± 13.93 | 1.60–9.70 5.21 ± 3.46 |
| Dry | 0.23–3.51 2.11 ± 1.03 | 0.54–56.07 21.17 ± 20.56 | 0.05–3.14 1.72 ± 1.21 | 1.39–2.56 1.84 ± 0.42 | 0.80–8.50 4.49 ± 2.99 |
| Windy | 0.81–1.03 0.93 ± 0.07 | 1.53–7.14 4.34 ± 1.83 | 4.32–8.86 6.24 ± 1.43 | 12.96–67.63 29.90 ± 20.69 | 1.50–10.70 4.70 ± 3.56 |
| Differences between seasons | F = 4.45 $p < 0.005$ | F = 7.04 $p < 0.005$ | F = 5.31 $p < 0.005$ | F = 11.25 $p < 0.005$ | F = 0.09 $p > 0.005$ |
| * Upper limits established for marine coastal waters | 0.002 | 0.04 | 0.01 | 5 | |

* Water quality criteria for the protection of aquatic life [25].

### 3.3. Major Phytoplankton Groups

The total average phytoplankton abundance was $5.3 \times 10^5$ cells L$^{-1}$, with a minimum value of $4.1 \times 10^4$ cells L$^{-1}$ (March) and a maximum of $8.8 \times 10^6$ cells L$^{-1}$ (May). Nanoflagellates varied slightly in abundance, decreasing in the dry season ($2.2 \times 10^4 \pm 2.8 \times 10^4$ cells L$^{-1}$) and increasing in the windy season with a mean value ± standard deviation of $8.9 \times 10^4 \pm 4.7 \times 10^4$ cells L$^{-1}$ (Table 2; Figure 2a). Significant differences occurred between seasons (F = 4.92; $p < 0.05$). During the rainy season, diatoms presented the highest abundances with a mean value ± standard deviation (SD) of $4.1 \times 10^5 \pm 3.4 \times 10^5$ cells L$^{-1}$, while during the windy season, the lowest abundances were recorded with a mean value ± SD of $3.6 \times 10^5 \pm 3.6 \times 10^5$ cells L$^{-1}$ (Table 2; Figure 2b). No significant difference between seasons was observed (F = 0.06; $p > 0.05$). The dinoflagellates showed the lowest abundances during the rainy season ($5.8 \times 10^4 \pm 5.4 \times 10^4$ cells L$^{-1}$) and abundances on the order of $10^4$ to $10^5$ cells in the rainy and dry seasons.

The highest abundance of dinoflagellates ($8.0 \times 10^4 \pm 1.3 \times 10^5$ cells L$^{-1}$) was observed in the dry season (Table 2; Figure 2c). No significant difference between seasons was observed (F = 0.11; $p > 0.05$). Cyanobacterial abundance notably increased in the rainy season, with an abundance of $4.0 \times 10^4 \pm 3.1 \times 10^4$ cells L$^{-1}$. Cyanobacteria showed total abundances of the same order of magnitude ($10^4$ cells L$^{-1}$) in all seasons (Table 2; Figure 2d). No significant difference between seasons was observed (F = 0.18; $p > 0.05$).

**Table 2.** Statistics for meteorological seasons for abundances of major phytoplankton groups (cells $L^{-1}$). Cell abundances obtained from the period 2019–2020 are presented as range and mean $\pm$ SD.

| Season | Nanoflagellates | Diatoms | Dinoflagellates | Cyanobacteria |
|---|---|---|---|---|
| Rainy | $7.0 \times 10^3 - 1.1 \times 10^5$ $4.3 \times 10^4 \pm 4.3 \times 10^4$ | $1.1 \times 10^5 - 1.0 \times 10^6$ $4.1 \times 10^5 \pm 3.4 \times 10^5$ | $1.1 \times 10^4 - 1.6 \times 10^5$ $5.8 \times 10^4 \pm 5.4 \times 10^4$ | $5.0 \times 10^3 - 9.8 \times 10^4$ $4.0 \times 10^4 \pm 3.1 \times 10^4$ |
| Dry | $5.0 \times 10^3 - 8.6 \times 10^4$ $2.2 \times 10^4 \pm 2.8 \times 10^4$ | $6.3 \times 10^3 - 1.5 \times 10^6$ $3.3 \times 10^5 \pm 5.4 \times 10^5$ | $1.4 \times 10^3 - 3.7 \times 10^5$ $8.0 \times 10^4 \pm 1.3 \times 10^5$ | $4.4 \times 10^3 - 9.0 \times 10^4$ $3.6 \times 10^4 \pm 3.3 \times 10^4$ |
| Windy | $2.3 \times 10^4 - 1.6 \times 10^5$ $8.9 \times 10^4 \pm 4.7 \times 10^4$ | $1.3 \times 10^5 - 1.1 \times 10^6$ $3.6 \times 10^5 \pm 3.6 \times 10^5$ | $2.0 \times 10^4 - 1.6 \times 10^5$ $7.4 \times 10^4 \pm 5.3 \times 10^4$ | $1.7 \times 10^4 - 6.2 \times 10^4$ $3.1 \times 10^4 \pm 1.6 \times 10^4$ |
| Differences between seasons | F = 4.92, $p < 0.005$ | F = 0.06, $p > 0.005$ | F = 0.11, $p > 0.005$ | F = 0.18, $p > 0.005$ |

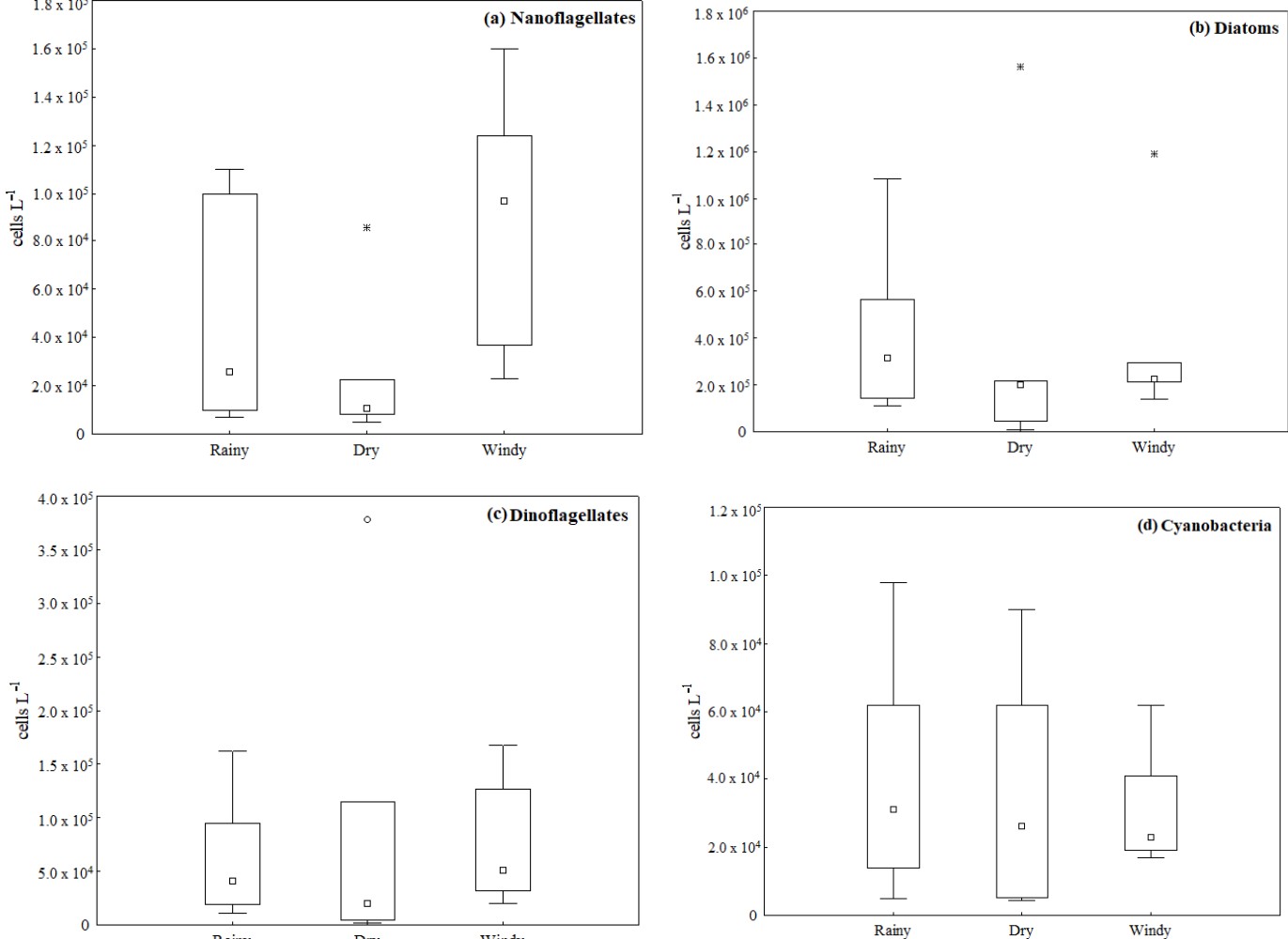

**Figure 2.** Abundances of major phytoplankton groups in three seasons. Cell abundances obtained from the period 2019–2020 (n = 21 for each box plot). Box plots indicate the range (vertical bars), first quartile, median (square) and third quartile. The outliers are indicated with a circle, and the extreme value is indicated with an asterisk (*).

*3.4. Relationship between Physical–Chemical Variables, Inorganic Nutrients and Phytoplankton*

The response of the major phytoplankton groups to physical–chemical variables and inorganic nutrients is primarily explained by the first two axes (Figure 3; axis 1: 56.8%, axis 2: 31.9%, total 88.7%). The correlation between major phytoplankton groups and physical–

chemical variables was high (r ≈ 0.8), indicating a significant relationship. Only the first canonical axis was statistically significant (F = 2.82, *p* < 0.05, Monte Carlo). However, the CCA graphs showed that the first axis (positive values) separated the samples of diatoms by higher values of pH and silicate and separated the samples of cyanobacteria with high values of temperature from the samples with dinoflagellates and nanoflagellates. Regarding the second axis, nanoflagellates were abundant in negative coordinates, coinciding with the samples with high values of ammonium, phosphate DO and DO%.

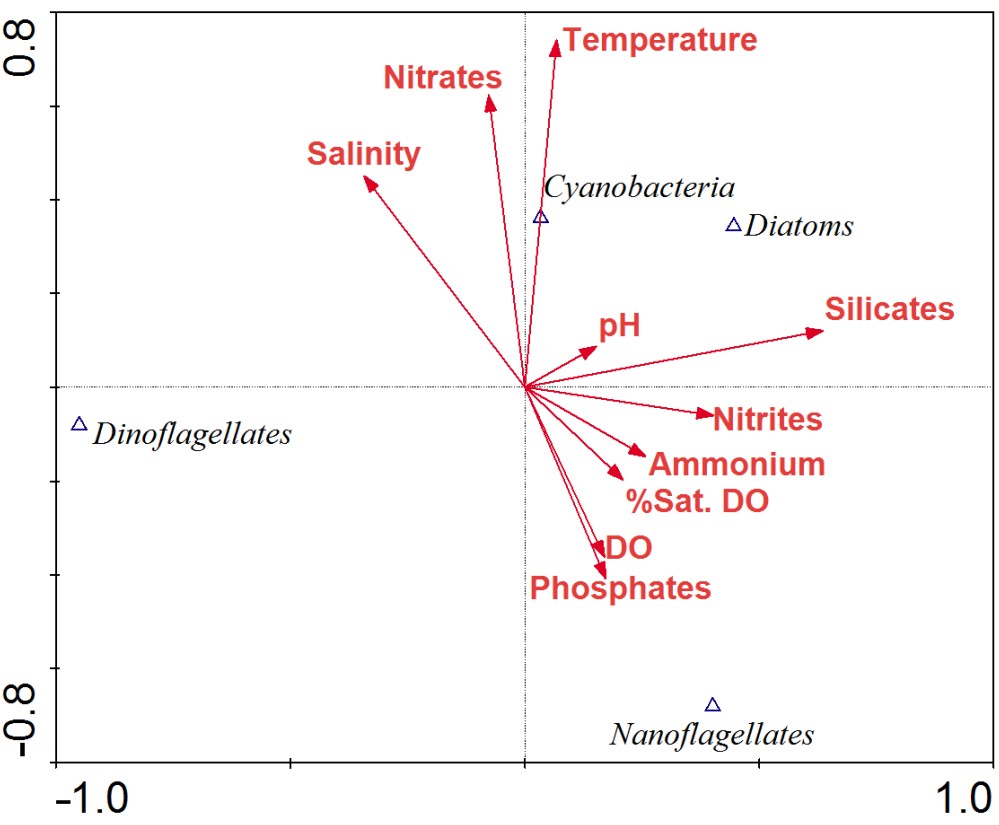

**Figure 3.** Relationships between major phytoplankton groups, physical–chemical variables and inorganic nutrients by CCA analysis. Data obtained from the period 2019–2020.

### 4. Discussion

Studies focused on the interpretation of the phytoplankton structure in maritime Mexican states are scarce; however, local reports of the potentially noxious species are more common. This results from the necessity of enriching our knowledge of selected harmful algal species in diverse coastal water bodies, seasonality and their occurrence, as these species are under the jurisdiction of the health and sanitary authorities. Along the northern Yucatán coast, the structure of the phytoplankton community is related to the discharge of nutrients, hydrographic conditions, turbulence and human impact [26].

In the northeastern part of the Campeche coast, along the western coast of the Yucatán Peninsula, the highest inorganic nutrient and chlorophyll-*a* values were due to the continental water runoff [27]. Based on chemical, phytoplankton and submerged vegetation variables obtained during more than 10 years of monitoring along the western coast of the Yucatán Peninsula, corresponding to the study zone of the present article, the level of eutrophication in the area of S1 (near Celestún; Figure 1) was characterized as moderate, and the area near S4 (near Seybaplaya) as bad [28]. Among three Mexican states (Campeche, Yucatán and Quintana Roo) to which the coasts of the Yucatán Peninsula belong, the marine coastal phytoplankton of Campeche exhibits greater cell abundance, species richness, equitability and diversity [28].

The physical–chemical variables (temperature, salinity and pH) recorded are typical for the central coast of Campeche, with seasonal characteristics influenced by the shallowness of the study area [29–31], while DO levels are subjected to the movement in the water body caused by rain and wind coupled with the action of photosynthesis carried out by photoautotrophic organisms [32].

The variation of inorganic nutrient concentrations is likely to be related to specific polluting activities, such as electricity production and aquaculture [33]; however, the main origins of inorganic compounds are urban wastewater discharges without prior treatment [30,31,34,35]. This is confirmed by the high registered concentrations of nitrogenous compounds that coincide with what has been previously reported in [30,34]. The entry of inorganic compounds produces a change in the concentrations of nutrients in water bodies, which affects nutrient chemistry and consequently biotic interactions and biodiversity responses [36].

An example of the previous background lies in the current condition of the study area, which may be contributing to changes in the taxonomic composition and abundance of the main phytoplankton groups. The dominance of diatoms was observed with its maximum abundances of up to $10^6$ cells $L^{-1}$ in the rainy season, contrary to what was reported earlier [30,31]. Evidence suggests the importance of nitrites in the metabolism and rapid growth of diatom cells [37,38]. This can be seen in the present study where nitrite presented maximum values, ranging from 0.38 to 3.24 mg $L^{-1}$ in the rainy season, reinforced by the high values of silicate (1.60–9.70 mg $L^{-1}$) essential for the optimal development of diatoms [39].

In the Gulf of Mexico, the development of major phytoplankton groups depends on the study region and its hydrographic conditions. For example, at the continental margin along the northern gulf, the river outflows, together with other environmental variables such as wind forcing and stratification of the water column, have a pronounced effect on phytoplankton [40]. Along the Louisiana coast and the adjacent part of the Texas coast, the phytoplankton species composition is related to meteorological seasons, physical parameters and nutrients that influence the total phytoplankton biomass [41].

Nanoflagellates are linked to high concentrations of reduced nitrogenous compounds [39] and are favored by low temperatures and salinities as shown by CCA. However, not all nanoflagellates prefer reduced forms of nitrogen [2,42]. This size fraction has frequently been reported as abundant (of the order of $10^6$ cells $L^{-1}$) and the main component of phytoplankton, followed by diatoms and dinoflagellates [29–31,43–45]. However, in our study, information about autotrophic phytoflagellates is lacking. According to CCA, the dinoflagellates did not have any relationship with the studied environmental variables. This may suggest a marked dominance of diatoms over dinoflagellates thriving in poorer nutrient conditions [38]. Nevertheless, the observed dinoflagellate abundances were similar to those reported for the study area [29–31] and in other regions of the Gulf of Mexico in the pioneering studies [46–49] that related diatoms and dinoflagellates with the biomass and the productivity in the gulf [44,45]. These two taxonomic groups, apart from including potentially toxic species under the jurisdiction of the health and sanitary authorities of the State of Campeche, contribute $10^4$ to $10^6$ cells $L^{-1}$ in abundance during the rainy season [50,51].

Cyanobacteria presented abundances of the order of $10^5$ cells $L^{-1}$ in the dry and rainy seasons when temperatures were higher than 30 °C, coinciding with what has been recorded in tropical regions due to high temperatures [52]. Furthermore, when salinity diminishes, cyanobacteria replace other taxonomic groups in the water column, which can be considered indicative of eutrophication. This was observed in Términos Lagoon (Figure 1) and has been previously reported along the coast of Campeche [29–31,50,51,53].

Changes in the phytoplankton community are not axiomatic because of the complexity and variability of phytoplankton [54], which may result in being considered an ecological process due to the complexity and particularities of each ecosystem [55]. This was proved by the results obtained in this study. In addition, phytoplankton assemblages are exposed to

the synergistic effects of known (selective grazing, nutrients, light, salinity and competition) and unknown selective pressures [56].

## 5. Conclusions

The use of statistical tools for the study of the relationships between the phytoplankton community and the environmental variables must be accompanied by a broad knowledge of the hydrographic conditions, anthropogenic activities and particular geographical conditions in a given coastal area. This will contribute to improving the statistical results by considering them in a broader scenario.

**Author Contributions:** Conceptualization, J.A.G.-F., J.R.-v.O., Y.B.O. and C.A.P.-D.; data curation, J.A.G.-F., J.R.-v.O., R.D.-C., Y.B.O. and C.A.P.-D.; formal analysis, J.A.G.-F., J.R.-v.O., R.D.-C., Y.B.O. and C.A.P.-D.; funding acquisition J.R.-v.O. and C.A.P.-D.; investigation, J.A.G.-F., J.R.-v.O., R.D.-C., Y.B.O. and C.A.P.-D.; methodology, J.A.G.-F., J.R.-v.O., R.D.-C., Y.B.O. and C.A.P.-D.; resources, J.A.G.-F., J.R.-v.O., R.D.-C., Y.B.O. and C.A.P.-D.; software, J.A.G.-F., J.R.-v.O., R.D.-C., Y.B.O. and C.A.P.-D.; visualization, C.A.P.-D.; writing—original draft, J.A.G.-F., J.R.-v.O., R.D.-C., Y.B.O. and C.A.P.-D.; writing—review and editing, J.A.G.-F., J.R.-v.O., R.D.-C., Y.B.O. and C.A.P.-D. All authors have read and agreed to the published version of the manuscript.

**Funding:** The financial support given to the project "Bioprospección de toxinas lipofílicas e hidrofílicas en fitoplancton marino y su presencia en peces comerciales del sureste del Golfo de México", 7825.20-PD, by the Gobierno del Estado de Campeche and the Tecnológico Nacional de México (2020–2021; project leader: C.A.P.-D.) is acknowledged. C.A.P.-D. appreciates the postdoctoral scholarship granted by CONACyT (Estancias Posdoctorales por México 2022). Anonymous reviewers kindly improved the manuscript.

**Institutional Review Board Statement:** Not applicable.

**Informed Consent Statement:** Not applicable.

**Data Availability Statement:** Not applicable.

**Acknowledgments:** We thank Marcia M. Gowing from Seattle, WA, USA, who kindly improved the writing style, J. Lopez-Ruiz (EPOMEX-UAC) and Natalia A. Okolodkova (Mexico City, Mexico) for the elaboration of the map, and the three anonymous reviewers for their valuable comments.

**Conflicts of Interest:** The authors declare no conflict of interest.

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
