# Peer review of "Seasonal Response of Major Phytoplankton Groups to Environmental Variables along the Campeche Coast, Southern Gulf of Mexico"

_phycology, doi:10.3390/phycology3020017_

Round 1

Reviewer 1 Report

Review for the paper "Seasonal response of major phytoplankton groups to environmental variables along the Campeche coast, southern Gulf of Mexico" by Juan Alfredo Gómez-Figueroa, Jaime Rendon-von Osten, Carlos Antonio Poot-Delgado, Ricardo Dzul-Caamal and Yuri B. Okolodkov submitted to "Phycology".

General comment.

Autotriphic micro-, nano- and picoplankton have been found to play several important roles in marine pelagic food webs, e.g. as primary producers thus regulating the efficiency of carbon cycling in aquatic ecosystems. The role of phytoplankton as producers has been extensively investigated over the previous decades, across a variety of marine habitats (estuarine to oceanic environments). Nonetheless, previous studies regarding phytoplankton diversity and succession have focused primarily on a certain season or event (e.g. algal bloom), and only a few recent studies have addressed this process over the annual cycle. Available data regarding phytoplankton dynamics over an annual time scale remain limited, particularly in temperate coastal waters, in which the levels of phytoplankton biomass and nutrient concentrations are expected to vary considerably in time and space. The authors conducted an annual survey to describe the seasonal cycle of phytoplankton abundance and composition in the shallow coastal waters of the Gulf of Mexico. They revealed different environmental impacts on diatoms and cyanobacteria. The study may be useful for monitoring of the coastal ecosystems in the region. Relevant methods to collect samples and to process the data were used in the study. The main results are visualized in a good way. Statistical methods are adequate and correctly used. After minor revision, this paper may be accepted for publication in "Phycology".

Specific remarks.

L20-21. Consider replacing "L -1 cells" with "cells L -1".

Material and methods. I suggest include a brief description regarding the climatic conditions in the study area as well as oceanography (water masses, circulation, tidal cycle).

L138. Consider replacing "cels L -1" with "cells L -1".

L148-149. Indicate what do mean vertical bars and points in the graphs.

L177. Consider replacing "premise" with "background".

Discussion. A short part comparing the authors’ data with similar sites would make the paper more interesting for a wider audience and I encourage the authors including this comparison in the ms.

Review for the paper "Seasonal response of major phytoplankton groups to environmental variables along the Campeche coast, southern Gulf of Mexico" by Juan Alfredo Gómez-Figueroa, Jaime Rendon-von Osten, Carlos Antonio Poot-Delgado, Ricardo Dzul-Caamal and Yuri B. Okolodkov submitted to "Phycology".

General comment.

Autotriphic micro-, nano- and picoplankton have been found to play several important roles in marine pelagic food webs, e.g. as primary producers thus regulating the efficiency of carbon cycling in aquatic ecosystems. The role of phytoplankton as producers has been extensively investigated over the previous decades, across a variety of marine habitats (estuarine to oceanic environments). Nonetheless, previous studies regarding phytoplankton diversity and succession have focused primarily on a certain season or event (e.g. algal bloom), and only a few recent studies have addressed this process over the annual cycle. Available data regarding phytoplankton dynamics over an annual time scale remain limited, particularly in temperate coastal waters, in which the levels of phytoplankton biomass and nutrient concentrations are expected to vary considerably in time and space. The authors conducted an annual survey to describe the seasonal cycle of phytoplankton abundance and composition in the shallow coastal waters of the Gulf of Mexico. They revealed different environmental impacts on diatoms and cyanobacteria. The study may be useful for monitoring of the coastal ecosystems in the region. Relevant methods to collect samples and to process the data were used in the study. The main results are visualized in a good way. Statistical methods are adequate and correctly used. After minor revision, this paper may be accepted for publication in "Phycology".

Specific remarks.

L20-21. Consider replacing "L -1 cells" with "cells L -1".

Material and methods. I suggest include a brief description regarding the climatic conditions in the study area as well as oceanography (water masses, circulation, tidal cycle).

L138. Consider replacing "cels L -1" with "cells L -1".

L148-149. Indicate what do mean vertical bars and points in the graphs.

L177. Consider replacing "premise" with "background".

Discussion. A short part comparing the authors’ data with similar sites would make the paper more interesting for a wider audience and I encourage the authors including this comparison in the ms.

Author Response

Specific remarks.
L20-21. Consider replacing "L -1 cells" with "cells L -1".

Answer: Done.

Material and methods. I suggest include a brief description regarding the climatic conditions in the study area as well as oceanography (water masses, circulation, tidal cycle).

Answer: Done.

L138. Consider replacing "cels L -1" with "cells L -1".

Answer: Done.

L148-149. Indicate what do mean vertical bars and points in the graphs.

Answer: Done (p. 5).

L177. Consider replacing "premise" with "background".

Answer: Done.

Discussion. A short part comparing the authors’ data with similar sites would make the paper more interesting for a wider audience and I encourage the authors including this comparison in the ms.

Answer: Done (pp. 6-8).

Reviewer 2 Report

This is a descriptive investigation of the algae inhabiting an aquatic environment. The authors did use up-to-date statistical analysis of the data including Correspondence Analysis. The data will be useful to protect the aquatic sites from man made perturbation. 

Author Response

We are grateful to Reviewer 2 for reading and evaluating our MS.

Reviewer 3 Report

I am deeply sorry to say that the paper needs some extensive improvement. Data presented are somewhat poor and discussion very superficial. I would recommend authors to improve their research in order to present something real interesting and scientificly important.-

Author Response

In regard to Reviewer 3, we regret that we could not address the criticisms in the absence
of specific details. We consider our research important, especially as a reference point and baseline for future studies. Although we would have liked to have had monthly samplings for higher resolution, our three samplings correspond to three meteorological seasons and therefore are of predictive value. In fact, our study represents a yearly baseline monitoring following a rather complete scheme of analysis of cell abundances of the major phytoplankton groups and the basic inorganic nutrients in time and space. In response to the detailed criticism of Reviewer 1, we improved the M&M and Discussion sections. Sampling at 7 points along the coast is one of our strongest points. Furthermore, identifying correlations between the major phytoplanktonic groups and a number of physical-chemical variables using CCA is another advantage of our results. All the
data are original, thus corresponding to a primary article. For comparison: most of the similar studies in the study region were limited to only specific taxonomic groups, whereas our study considered diatoms, dinoflagellates, cyanobacteria and nanoflagellates. Cell counting was performed using the internationally accepted Utermöhl ́s technique. We recognize that the monitoring scheme could be improved (clearly, if provided with much more financial support).
Nevertheless, our study is scientifically important, especially in such a poorly studied region with a significantly increasing anthropogenic impact. We should emphasize that in Latin-American countries in general, governmental support is relatively low, and data are frequently published in the so-called grey literature.
Our contribution intends to fill this gap.

Round 2

Reviewer 3 Report

Authors did not answer part of my questions posed before. One major question was regarding the collection is just one day per climatic season. Did the authors picked a very regular day in each season to do their collecting? Difficult to answer and difficult to believe. However, some improvement was detected, and I decided to recommend the publication of the article in "Phycologia".-